# Beyond Magic: Fostering Literacy Resilience in Diverse Classrooms through Home-Based Approaches

**DOI:** 10.3390/bs14090834

**Published:** 2024-09-18

**Authors:** Dolly Eliyahu-Levi

**Affiliations:** Language Department, The Multidisciplinary Faculty, Levinsky-Wingate Academic College, Tel Aviv 6937808, Israel; doly.levi@l-w.ac.il

**Keywords:** family literacy, literacy resilience, language teaching

## Abstract

The classrooms in Israel are very diverse, with students differing in learning styles, their handling of literacy tasks, personal and socioeconomic backgrounds, and more. These differences significantly impact the curriculum aimed at promoting literacy resilience, explicit teaching processes in the classroom, and imparting metacognitive strategies and actions to overcome learning difficulties. This qualitative-interpretative study reveals the pedagogical perceptions, challenges, and coping strategies of fourteen Hebrew teachers in five elementary schools in central Israel regarding integrating home literacy in language lessons and cultivating literacy resilience among their students. The research data were collected through in-depth interviews with the teachers. The analysis of the teachers’ reports reveals two main perceptions regarding literacy resilience: (1) Literacy resilience is a tool for life; (2) Home literacy significantly contributes to fostering literacy resilience. Furthermore, cultivating literacy resilience presents three significant challenges for the teachers: (1) Teaching in a heterogeneous classroom, (2) Encouraging parental involvement, and (3) Fostering independent learners. To cultivate literacy resilience in a heterogeneous classroom, teachers must be sensitive to each student’s unique needs and plan teaching-learning processes based on principles of self-directed learning and peer dialogue. They must establish a personal-emotional connection that is a significant anchor for the students and outlines a path for integrating and strengthening the sense of competence in handling literacy tasks. It was also found that parental involvement is a significant factor influencing the cultivation of literacy resilience, and teachers undertake various actions to increase their level of involvement. This study adds an essential layer to the body of knowledge regarding the understanding of the factors affecting the development of pedagogical literacy perceptions that promote the integration of home literacy in the classroom. These perceptions may promote the nurturing process of literacy resilience among students from various cultures, accepting and understanding them. In this way, we can attempt to address literacy and language challenges in Israel.

## 1. Introduction

Linguistic literacy is a critical component in the academic and personal development of students and significantly affects their success in their studies. It is a basis for cognitive, social, and professional development. Students with high linguistic literacy can understand complex texts, analyze information effectively, and express themselves clearly—essential skills in all areas of life. Linguistic literacy has implications for adult life, because it can help graduates integrate successfully into the world of work, continue to learn and develop throughout life, and actively participate in democratic society [1,2].

One of the main goals that educators face is the acquisition of linguistic literacy skills and mastery of literacy skills. In other words, literacy resilience refers to the ability to deal independently with a wide variety of linguistic tasks in different environments. This includes the ability to understand and produce different types of texts, use language in diverse contexts, adapt the linguistic style to different situations and cultivate critical thinking, identify difficulties and know how to overcome them through diverse and relevant metacognitive strategies and actions [3].

The challenge is extremely complex in a heterogeneous classroom where students with varying levels of linguistic literacy from different cultural and linguistic backgrounds learn together, each requiring an adapted approach. Educators’ time and energy resources are limited, yet they are required to plan differential lessons to foster literacy resilience and maintain a balance between advancing high-performing students while supporting others, all while creating interest and motivation [4].

One of the ways to address this challenge is to increase the involvement of family members in the process of fostering students’ literacy resilience in the classroom. Therefore, it is important to examine the perceptions and actions of Hebrew language teachers in elementary schools regarding the importance of integrating family literacy in the process of cultivating literacy resilience among their students.

## 2. Literature Review

### 2.1. Resilience

Resilience is not a genetic trait or traditional learning through memorization of theoretical material or scientific calculation, but rather a process of cultivating strong belief and the ability to be worthy and capable of dealing with challenges through expression of interaction, flexibility, and involvement. Resilience does not develop naturally in adulthood but must be nurtured from a young age [5,6]. Moreover, resilience is a mix of personal traits, processes, and mechanisms through which strengths are harnessed to cope with distress [7,8,9]. Similarly, Clauss-Ehlers [10] defined resilience as a process, ability, or outcome of successful adaptation despite challenging or threatening circumstances and noted that resilience from this perspective is critical to discussing the well-being of children and adolescents, as young people with these abilities can overcome distress and difficult life circumstances more than adults.

Examining resilience reveals that children are influenced and shape their perspective through the social world around them, the stories they tell themselves, and interactions or relationships with other children, parents, relatives, or teachers. Therefore, how a child responds in childhood can greatly impact adulthood. Children who have developed resilience can better cope with stresses, failures, disappointments, or emotional challenges. They can be flexible and adapt more quickly to changes at home, school, and life [11]. Examining resilience from a literacy perspective indicates that teachers who foster resilience among their students—optimism, social skills, problem-solving, decision-making, positive self-image, sense of belonging to family, school, learning, and society—can increase resilience and change the life course of children [6].

### 2.2. Resilience Pedagogy

Benard [12] proposes that Resilience Pedagogy is an approach that views enhancing and deepening resilience as an integral part of a teacher’s role within the teaching-learning process. He argues that one of the key responsibilities of educators is to build resilience in students, particularly adolescents, by developing their ability to integrate fragmented experiences resulting from failures or traumatic events. He emphasizes that to effectively foster resilience in their students, teachers must first cultivate their resilience. Both teachers and students who have developed resilience are better equipped to navigate uncertainties, unexpected situations, and social distress in their daily lives.

Research widely acknowledges that teachers are among the most influential factors in the lives of children from diverse social backgrounds [13]. Namka [14] highlights the crucial role schools play in children’s lives, often serving as their primary source of support. Schools provide a space where children receive attention, empathetic listening, and emotional support from their teachers. However, most educators tend to prioritize the didactic aspect of teaching, viewing it as an essential component and often focus on rewarding children for academic achievements or expected behavior. In contrast, the personal component that allows for flexibility and focuses on nurturing students’ strengths while promoting well-being and resilience is often perceived as a less critical and potentially less effective aspect of education [5,15].

### 2.3. Literacy Resilience

Literacy is the ability to identify, understand, express, create, and interpret concepts, feelings, facts, and opinions orally and in written forms, using visual, audio, and digital materials in various fields of knowledge and contexts [16]. It implies the ability to communicate effectively with others, appropriately and creatively. Linguistic literacy includes skills in reading, writing, correct understanding of written information, and the ability to formulate and express things orally in a convincing manner appropriate to the context [17]. In recent years, there has been a need to expand the definition of linguistic literacy, particularly reading literacy, and aspects of reading digital texts and meta-cognitive processes have been included [18].

Literacy resilience is the learner’s mastery of literacy skills: the ability to cope with literacy tasks, develop critical thinking, identify difficulties, and know how to overcome them using various relevant metacognitive strategies and actions. Literacy resilience does not develop spontaneously but requires explicit instruction and scaffolding [15]. Linguistic literacy is important in its contribution to academic achievement and optimal integration into the adult life of students. Students who struggle with literacy skills present low academic achievement, low self-perception, and fewer expectations for progress [19,20]. Researchers agree that the relationship between teacher and student was found in the literature as the most significant factor in nurturing literacy resilience and that the level of children’s achievement is largely a function of the quality of teaching [21]. Building resilience in school is accomplished by adapting the content taught, adapting teaching methods, and tightening the interpersonal connection between teacher and student. Children need to be made to develop positive thoughts of growth [22], develop intelligence and abilities [23], believe in themselves to be able to achieve personal and academic goals, cope with distress and overcome difficulties [24]. Therefore, it is important to invest resources in professional development related to literacy instruction, to empower teachers’ sense of self-efficacy and improve students’ learning processes and achievements [25,26].

Regarding teaching methods, most studies suggest using resilience principles as early as the teacher training stage [27]. Resilience pedagogy is expressed in the use of teaching methods that empower students and increase their resilience, including expanding student autonomy, encouraging critical thinking, and developing the ability to express personal opinion; in instilling messages of increasing capability and strengthening self-confidence within teacher–student relationships around teaching, learning and assessment; in consolidating the group of learners into a group that provides resilience and security to its members, and more.

The COVID-19 crisis and the transition to distance learning also deepened the literacy gaps in the heterogeneous classroom and increased the number of students struggling with reading, writing, and speaking. Literacy gaps have social implications, as low language skills can cause dropout from school, prevent social mobility, and weaken the country’s social cohesion. Despite exposure to sociodemographic risk factors, including poverty and low parental education, teachers can nurture literacy resilience among their students in heterogeneous classrooms, and one of the research questions is what actions are taken to nurture literacy resilience.

### 2.4. Family Literacy

Literacy development is generally perceived as inherently social, resulting from collaboration between the child and more experienced others, and family members play an important role in the development and mastery of their children’s literacy skills. Studies [28,29,30,31] prove that parents’ literacy level and how they encourage their children to understand and perform affects the establishment of children’s literacy resilience. Family interactions and productive relationships in the family unit can help in constructing literacy skills. Hence, parents and siblings nurture family-cultural literacy knowledge and skills that teachers can adopt and integrate during the lesson [32]. Family members play a central role in children’s literacy development, and they constitute a rich social, cultural, and linguistic resource that is often untapped by the education system [33]. Educators often find it challenging to leverage the literacy resources available in students’ homes, particularly when working with families from low socioeconomic backgrounds who are marginalized socially. Parents may lack the knowledge, time, energy, language skills, transportation, flexible work schedules, or social support needed to effectively engage with their children’s learning challenges [34,35].

Therefore, it is important to examine the perceptions and actions of Hebrew teachers in elementary school regarding the importance of integrating family literacy in the process of nurturing literacy resilience among their students. This integration can strengthen the connection between home and school and promote cooperation and parental involvement. It is recommended already in the training process to raise the level of awareness among student teachers to recognize the literacy knowledge of families as a rich source of information that can assist in developing a pedagogical perception that promotes actions to characterize family literacy and plan teaching methods in the classroom in relation to them [36].

Nurturing literacy resilience and integrating home literacy shapes the teaching-learning experience in a heterogeneous classroom and directly contributes to establishing teaching methods based on critical awareness about integrating home literacy in the classroom while showing openness, curiosity, empathy, and release from labeling and stereotypes. The research is relevant to a global reality where teaching-learning processes in heterogeneous classrooms are on the pedagogical agenda in Israel and other countries in the world dealing with waves of immigration and processes of integration and inclusion. This research will add an important layer to the body of knowledge regarding understanding the factors influencing the cultivation of literacy pedagogical perceptions and understanding that promote the integration of home literacy in the classroom. These perceptions may promote a process of nurturing literacy resilience among students belonging to various cultures, accepting and understanding them. In this way, we can try to deal with the challenges of integration in Israel.

The purpose of the research is to examine educational perceptions and pedagogical actions of Hebrew teachers in elementary school regarding nurturing literacy resilience and integrating home literacy in language lessons in the classroom. The research questions are:(1)What are the educational perceptions of Hebrew teachers in elementary school regarding nurturing literacy resilience and integrating home literacy in language lessons in the classroom?(2)What are the challenges and coping methods adopted by Hebrew teachers in elementary school to nurture literacy resilience and integrate home literacy in language lessons in the classroom?

## 3. Methodology

### Research Method

This research is a qualitative-interpretive study combining description, analysis, interpretation, and understanding. The focus of the interpretive paradigm is on understanding the complex world of experiences of Hebrew teachers in elementary school from their perspective, with a holistic view of processes occurring in the classroom and home literacy space [37]. According to this approach, the researcher observed the teachers and the pedagogical discourse as they are, during their natural occurrence, without attempting to manipulate them [38,39]. The research method allows hearing the personal voice of the teachers concerning the process of nurturing literacy resilience based also on home literacy and formulating actions to integrate it in the classroom. This approach may help in adapting a curriculum that will allow for nurturing literacy resilience, developing intercultural competence, openness, mental flexibility, respect, and inclusion of students from other cultures [40].

Fourteen Hebrew women teachers from five elementary schools belonging to the public stream in central Israel participated in the study. All are native Hebrew speakers, with a bachelor’s degree and teaching certificate received from teaching colleges. The teachers have three to five years of experience in teaching and classroom education in schools. Ten of them work full-time in schools, and four work part-time. None of them hold additional roles in the school.

To address the research questions, semi-structured interviews were conducted to efficiently engage participants in conversation and elicit their understanding and interpretation of a topic [41]. The interview was conducted face-to-face or by phone, lasting about 45 min each. Here are examples of questions asked in the interview: Describe how home literacy can be integrated into teaching-learning processes. To what extent do you think family literacy affects students’ literacy resilience? What are the difficulties you encountered during language lessons? How did you overcome them? Tell about the nature of the relationship with parents and the importance of its contribution. Describe the feelings and reactions of students in a lesson where you integrate home literacy.

The interviews were analyzed using an interpretive approach through content analysis focusing on what the teachers said in words, descriptions, and their place in the way they presented their words [42]. Krippendorff [43] distinguishes between different types of sampling and documentation units in the content analysis procedure. One of the relevant types for this research is thematic units defined by text characteristics, and their definition involves human judgment. The distinction between thematic units and the choice of which to sample for the research was made based on theoretical considerations, and in many cases are set against the rest of the findings that are not relevant to the research. According to Shkedi [44], content analysis is like a window allowing a view into the inner experience reflecting perceptions and actions in the classroom and school space.

Content analysis facilitates an accurate data description and allows for drawing valid conclusions within a broader context. The process of coding and classifying findings is a cyclical task that bridges theory and content, requiring the researcher to dedicate time for reading, probing, refining, and decision-making.

The content analysis process consists of three main stages: (1) Open Coding: This initial stage involves defining the first categories of analysis. The researcher searches for recurring phrases and ideas, uncovering a set of topics and organizing them into central categories that reflect teachers’ perceptions and pedagogical actions regarding the cultivation of literacy resilience and the integration of home literacy in Hebrew lessons. (2) Axial Coding: This stage involves rereading the texts to identify and refine the main thematic units. The researcher aims to tighten and determine which thematic unit each section belongs to, guided by the research goals and the theoretical conceptual framework that embodies the research objective. This framework focuses on cultivating literacy resilience in diverse classrooms while considering home literacy. (3) Selective Coding: In this final stage, the researcher determines recurring themes that will serve as central themes and adapts the segments to these themes. The goal is to develop a system of categories that provides a meaningful structure to the collected data. This approach aims to create a robust analytical framework that aligns with the research objectives and provides insightful interpretations of the data [45,46].

This research received approval from the ethics committee of the academic institution—approval number: 202350201.

## 4. Findings

The analysis of the findings revealed two main themes: (1) Perceptions—literacy resilience, integration of home literacy; (2) Challenges and coping methods—teaching in a heterogeneous classroom, encouraging parental involvement, and nurturing independent learners.

### 4.1. Perceptions

Examination of the educational perceptions of Hebrew teachers in elementary school regarding nurturing literacy resilience and integrating home literacy in language lessons in the classroom revealed two main perceptions:

#### 4.1.1. Literacy Resilience Is a Tool for Life

Literacy resilience means resilience, mainly being independent in learning, and dealing with various difficulties and challenges in their learning process even if these are skills they do not necessarily know, and from their knowledge, they know how to compensate for the difficulty, now, during the lesson and in life in general. We invest many resources in developing language skills and nurturing resilience, providing diverse opportunities for developing literacy skills.(Teacher S)

Literacy resilience is the student’s ability to deal with language difficulties in real time even when there are unfamiliar elements. That means not panicking from uncertainty, to contain, to deal with these difficulties. He doesn’t let the difficulty of any kind manage him, but he manages the difficulty. Children with literacy resilience are active in class, independent and not afraid to make mistakes. These are tools for life.(Teacher T)

S. believes that independent learning allows students to uncover new skills, formulate new knowledge, cope with challenges, and compensate for difficulties using knowledge and strategies they acquired in the past. It seems that such learning processes are based on inquiry and discovery, which increase motivation and personal involvement. Teacher T. emphasizes coping with language challenges and expands on the ability to better deal with pressures, failures, and difficulties in the learning process with self-confidence, understanding, choice, and without fear of making mistakes or feeling coercion from the teacher. Both teachers reveal a perception that literacy resilience is personal, characterizes a student with high awareness of what is happening in the learning process, takes responsibility, makes informed choices, and acquires skills that will make them a lifelong learner: “They know how to compensate for the difficulty, now, during the lesson and in life in general”.

Studies indicate that a pedagogical setting based on synchronous and asynchronous digital tools can foster independent learning skills and promote active learning by encouraging learners to take responsibility for their education. This involves making informed choices that require a high level of awareness and control over the learning process [47]. Furthermore, motivation for autonomous learning is linked to positive self-experiences, fostering a sense of optimality and promoting optimal functioning [48].

The reports also show: “We invest a lot of resources in developing language skills and nurturing resilience” that teachers identify the relevance of literacy resilience to students’ lives in the long term, and they are aware of the facts: (1) that resilience is not static and innate, but has a flexible character and can be nurtured; (2) that students with high literacy resilience tend to be more independent, good, efficient students with higher achievements. Therefore, they offer students experiences related to literacy and act to establish a variety of literacy resilience anchors.

#### 4.1.2. Home Literacy Has an Important Contribution to Nurturing Literacy Resilience

Building literacy resilience is not the responsibility of the educational framework alone, but a process that requires understanding and cooperation between the classroom, school, and home:

The main literacy resilience should also come from home, from the parents. It’s important to emphasize language at home from a very young age, like reading books, it’s a slow, long and prolonged process. Parents who know how to push and encourage, see the literacy development of the child. There are many students I see whose parents themselves do not know Hebrew well, and can’t help, and then the student progresses more slowly. I can’t take responsibility for the parents too.(Teacher B)

Parents need to be involved, in working with children on topics we learned in class. Parents who speak with a high vocabulary, enrich the language in speech, reading, and writing, are interested in what the child is learning, encourage them to continue practicing at home, and strengthen literacy.(Teacher N)

In relation to the teachers’ reports, building resilience is a slow, long and prolonged process, requiring interaction and cooperation between teachers and parents aimed at creating a good and worthy foundation for nurturing literacy resilience. The cooperation between the classroom space and the home space is changing, dynamic and adapted to the abilities of parents and children. Through it, children can be encouraged, language can be enriched in speech, reading, writing, give them a sense of progress and success, and help children deepen their language and speech skills.

Teacher B. creates a link between parents’ ability to help and the rate of children’s progress: “Parents who know how to push and encourage, each according to their ability, see the literacy development of the child. There are many students I see whose parents do not know how to help, and then the student progresses more slowly” B.’s words illustrate that nurturing literacy resilience occurs in a network of personal-social relationships intertwined with each other, and the child’s rate of progress is an expression of the parents’ ability. It seems that to understand and assess the literacy development of the child, we need to examine the interactions at home and with parents.

From a critical point of view, it seems that teachers do not believe they can nurture literacy resilience solely based on classroom learning activities while addressing diversity and variation in the classroom, and therefore repeatedly emphasize the importance of parental involvement and learning activities at home. However, teachers may be captive to stigmas and stereotypes about diverse families, and avoid, consciously or unconsciously, deepening their familiarity with the student’s sociocultural background. In doing so, they may be preventing their students from overcoming language gaps and fostering literacy resilience. Shmuel [49] found that teachers in the public education system in Israel are still not sufficiently aware of the importance of acquiring a deep understanding of their students’ historical and cultural background, and still fail to build a true partnership with parents.

From this finding, it is possible to draw an analogy to studies that examined the involvement of parents in their children’s learning process during the COVID-19 period. These studies found that the pandemic increased the need for frequent communication between teachers, parents, and students, as well as heightened the involvement of parents, family members, and the community [50,51,52]. Other studies [53,54,55] revealed that, from the parents’ perspective, they enriched their understanding of learning processes, had the opportunity to assist their children in completing academic tasks, played a more significant role in their children’s education, and developed a greater appreciation for the work of teachers. Similarly, a Portuguese study [56] demonstrated that parents adopted new interaction patterns to support their children, offering guidance, assistance, and continuous monitoring of educational task performance.

Teachers are expected to recognize the different learning needs of their students, have a broad pedagogical knowledge base, and adapt teaching-learning methods to the diversity of students in the classroom. To lead all students to meet the required achievements and standards in order to realize equality—all to ensure the quality knowledge base: academic, intrapersonal, and interpersonal needed to function in society.

### 4.2. Challenges and Coping Strategies

#### 4.2.1. The Challenge—Teaching in a Heterogeneous Classroom

Managing a heterogeneous classroom, where students with different levels of knowledge, abilities, and cultural and linguistic backgrounds learn together, requires diverse teaching skills from the teacher to address the unique needs of the learners:

One of the challenges in language lessons is reading acquisition in a class where each student is at a different level. Each student has their difficulty, and for each student, you need to adopt the appropriate method to acquire the language. It’s not always possible, there’s no time, many tasks and requirements, no help. I’m not a magician.(Teacher S)

Teaching in the classroom is very complex, the teacher is expected to adapt to each of the students, so that everyone receives help, feels success, and progress. No one feels frustrated, there are a lot of emotional aspects here, and we can’t settle for our limited hours in full classes to advance them.(Teacher T)

Teaching in a heterogeneous classroom is not new in the educational field. However, teachers perceive diversity as a challenge, emphasizing the many differences between children, and expressing doubt about whether it is indeed possible to foster literacy resilience and reduce gaps and inequality in a heterogeneous classroom under existing conditions: “It’s not always possible, there’s no time, many tasks and requirements, no help”. S’s choice of words “I’m not a magician” demonstrates her inability to perform tricks, illusions, or sleight of hand in a heterogeneous classroom. To address diversity under pressure and without help.

The repetition of the phrase every + student: “every student, for every student, for each of the students, that everyone” illustrates the fact that the whole class is composed of many individuals, and to all of them without exception, the teacher is required to provide a tailored response, and to react practically to diversity while adapting teaching methods to the cognitive and literacy abilities of the children: “For every student, you need to adapt the appropriate method for them to acquire the language”.

Teacher T openly criticizes the scope of Hebrew teaching hours in elementary school and the large number of students in the class: “We can’t settle for our limited hours in full classes to advance them.” It seems that N is aware of the Israeli reality where the Ministry in Israel poses a complex challenge for teachers to provide a tailored response in teaching Hebrew in heterogeneous and crowded classrooms. Studies [57,58] indicate that in classrooms, differences are found between students in readiness for learning, areas of interest, learning styles, cognitive abilities, experiences, personal-cultural background, and social status. However, teachers are expected to bear educational responsibility to respond to diversity while realizing the principle of equality.

Here are two reports illustrating how teachers cope with the challenge of teaching in a heterogeneous classroom:

In principle, I deal with diversity and heterogeneity in teaching methods; I try to adapt a learning method for each child even if it’s different from the usual. I do not see it as a difficulty. First, the child feels a sense of enjoyment, and the material sits well in memory—cards, pictures for illustration, and games.(Teacher D)

There are 32 children in the class, and we need to adjust for everyone. Everyone is involved, and everyone is truly active and feels confident to take part in the lesson despite the different levels. I work in small groups, some next to me, some in the space. A lot and regular individual meetings that strengthen the children also emotionally.(Teacher A)

Both recognize the diversity among students and reveal a differential learning-social perception, allowing all students to participate in the lesson. Through the differential response, D tries to strengthen each of the students where they are, improve their achievements, and allow them to experience success. Their words: “I try to adapt a learning method for each child” and “We need to make adjustments for everyone” illustrate their way of dealing with heterogeneity while striving for optimization of learning and equality in providing opportunities that allow each student to find a place in the classroom and reach maximum improvement of their achievements.

A regularly conducts groups and individual meetings that strengthen students’ functioning in emotional, social, academic, and cultural aspects. The individual or group personal meeting is an opportunity to conduct a learning-social interaction and integrate positive relationships between the friends in the meeting. This type of learning can create enjoyment and promote achievements. The report on integrating the “with me-next to me-in space” teaching model is reinforced by another study [59], which found that teaching adapted to the student’s skills and learning style and incorporating teacher mediation allows the student to take responsibility for the learning process and be goal-oriented. Implementing the model allows for close and caring relationships between the teacher and her students. Researchers in the global scientific community [60,61] believe that addressing emotional-social aspects can positively influence academic achievements in the present and cultivate the student to be a responsible citizen in a democratic state based on the values of equality, human dignity, and social justice.

#### 4.2.2. The Challenge—Encouraging Parental Involvement

Encouraging parental involvement in a heterogeneous classroom requires effective communication from teachers, proactive actions, knowledge sharing, and fruitful cooperation to foster literacy resilience and promote academic achievements:

Parents need to understand that children are in the process of reading acquisition and that it needs to be done and worked on. I give homework assignments many times, and the children who do them are those whose parents are more involved, and therefore they progress more.(Teacher A)

Parents do not always understand how important it is to read and practice. One way is to give assignments that also involve the home and involve the family. Like research tasks that require parental cooperation, students ask their parents about the family. In this way, we can also present the home inside the classroom and strengthen the connection.(Teacher T)

Family literacy and parental involvement in children’s reading and writing have an emotional impact and strengthen the child’s confidence. Collaboration between parents and child is enjoyable and increases motivation to delve into the content itself and understand it better. I opened a WhatsApp group for parents; I uploaded instructional videos with my explanations, and parents asked questions and shared ideas.(Teacher R)

All three teachers testify that raising parents’ awareness of the importance of their involvement in fostering literacy resilience is critical to ensure that students receive maximum support at home. According to them, parents are the most important partners, constitute a rich social-linguistic resource, and have an important role in the development and mastery of their children’s literacy skills.

A’s words: “The children who do them are those whose parents are more involved, and therefore they progress more” teach that there is a connection between parental involvement and student achievement and fostering literacy resilience. In other words, parents who cooperate with the educational staff, are involved in homework preparation, provide emotional and academic support, and assist or positively influence their children’s achievements in the literacy field. The basis for this finding is found in other studies [56,62,63] that examined interactions between teachers, parents, and students and found that in learning processes where parents are involved, students enjoyed academic and emotional support and strengthening of self-efficacy in learning.

Teacher T emphasizes in her report the importance of the connection between the classroom and home, and notes that joint assignments for parents and children are a way to increase parental cooperation and involvement: “Research tasks that require parental cooperation”. Exposing the home space will allow teachers to get to know the students from a personal-family perspective, demonstrating that the exposed family knowledge is significant for them, serves as a source for improving spoken and written discourse abilities in the classroom, and allows them to give a more precise response to their diverse needs. Moreover, information about home culture invites multi-cultural discourse on similarities and differences to recognize the value of the individual and out of aspiration to give place in classroom discourse to values, beliefs, and traditions of different cultures. Exposure to diverse voices can preserve the unique place for each one, empower a sense of mutual enrichment and at the same time create a collective.

One way to encourage parental involvement is through a collaborative WhatsApp group that serves as a digital platform for ongoing support and not just in one-time events: “I upload instructional videos with my explanations, parents ask questions and share ideas”. The group allows maintaining continuous contact at times convenient for parents without the need to leave home or miss work hours. This way, even busy parents can be involved in their children’s education. Also, the digital platform is suitable for parents who do not speak Hebrew or come from diverse cultural backgrounds. Through the WhatsApp group, translation and explanations can be provided in different languages, visual tools can be used, and content can be adapted to diverse cultures. This type of communication allows parents to be involved despite language and cultural barriers. It appears that the educational setting lacks a robust technological-digital infrastructure necessary for the effective operation of a virtual learning campus through platforms like Moodle or Canvas. Evidence suggests that many educational institutions face significant challenges in implementing digital learning, primarily due to concerns regarding its effectiveness and the capacity of both students and educators to adapt to new digital learning environments. A major challenge is the requirement for high levels of self-discipline among learners, coupled with existing gaps in digital literacy skills and infrastructure [64,65].

It seems that Teacher R acts to establish close and ongoing relationships with parents while revealing commitment from both sides. She also conceptualizes literacy knowledge for parents and tries to encourage them to increase their involvement. The personal connection allows for a dialogue where teachers learn to recognize the strengths and weaknesses of parents, adapt the educational response, and increase parents’ sense of capability and involvement in the educational process.

### 4.3. The Challenge—Cultivating an Independent Learner

Independent learning allows learners to take responsibility for their learning process, develop thinking skills, and flexibility in learning. However, it also presents challenges that require self-discipline and appropriate support. Cultivating an independent learner in elementary school is a unique challenge as it involves young ages where students are still developing basic skills and learning abilities:

Teaching children at home and in the classroom to be independent, they can perform reading comprehension tasks on their own from start to finish. Explaining to them that I do not immediately approach to answer them when they ask, because it’s important that they will try on their own. To reduce my mediation to a minimum according to the student’s level. Some children do not believe they are capable; they sit and wait for me.(Teacher N)

Teacher N’s report reveals the connection between self-efficacy and the ability for independent learning. Her words: “There are children who do not believe they are capable” indicate that a sense of self-efficacy and self-confidence are essential components in cultivating an independent learner, and they affect motivation and the ability to cope independently with the literacy challenge. It seems that some students lack self-confidence, and do not believe in their ability to succeed in the task and experience success. Here’s another report:

I teach in plenary and give independent practice on the task. Not everyone is capable; some do not have learning habits to sit, read, try, or make an effort. There’s no motivation and desire to succeed, even when the task is close to their world, or they choose themselves. They are used to coming to the teacher and getting answers. This is not the way.(Teacher A)

Teacher A reports that some students are not accustomed to independent learning. They do not learn out of interest and curiosity, do not make an effort, and learning for them may become a chore instead of an experience. A is aware of the fact that in order to arouse students’ curiosity and motivation, she needs to incorporate materials and tasks from the students’ world outside the school framework and allow them to choose. But even then, they do not always mobilize their self-efficacy and choose to come ask the teacher. Hattie [66] suggests that teachers enrich their knowledge about their students’ needs and preferences, thereby increasing the successful experience of the lesson among their students and enriching their pedagogical toolbox.

One way to cultivate independent learning skills is through dialogic peer learning, which emphasizes interaction between learners, mutual understanding, and the process of creating collaborative knowledge meaning. Learners listen to each other, help each other express their ideas, and search for answers and explanations. Researchers believe [67,68] that interaction in pairs or small groups is more suitable for this than discussions involving the whole class. Here’s a report from Teacher B:

It’s important to allow them to work together, think together, and explore. This mainly helps students who wouldn’t do anything alone, and it strengthens their confidence. I hear them talking, raising ideas, helping and explaining to each other, and less turning to me. Slowly friendships are formed, and they are more open.(Teacher B)

The dialogue and interaction between students force them to be more involved and active while taking responsibility and relying less on the teacher. The teacher is in the position of an advisor who assists when needed and avoids giving specific instructions regarding how to perform the activity. It also emerges that interactions involving a joint task affect social connections, openness, and establishing trust between students so that each of them is aware of themselves and others around them. Researchers [69] believe that collaborative-dialogic learning occurs thanks to the teacher’s interaction with learners and helps cultivate independent learning skills.

Another way to deal with the challenge of independent learning is through building routines of personal conversation and opening opportunities for learning:

I see the gaps between the children; the personal conversation allows me to get to know the students to adapt the learning and support to them.(Teacher T)

It’s important to set personal conversations, establish a personal connection with the students, and learn what interests or bothers them. To tell them that I’m here to help and explain what I expect in terms of their behavior in class and how they deal with tasks.(Teacher D)

Personal conversations are significant moments that provide an opportunity to establish a sense of belonging and care and to be a significant adult figure for the student. Both teachers report that personal conversations are an opportunity for a deeper acquaintance with the student. This acquaintance includes understanding their areas of interest, strengths, weaknesses, and aspirations. It’s possible that when a student feels that the teacher is interested in them as a person and not just as a student, they tend to be more open to share their thoughts, feelings, and difficulties they experience. All these allow the teacher to adapt the teaching to the personal needs of each student and create a personalized learning environment in a way that will lead to academic, emotional, and social growth. This finding is related to other studies [15,70] that examined the contribution of personal connection to promoting learning and increasing sense of capability and found that establishing close relationships of listening out of concern also for personal problems helps in dealing with emotional-social tensions. Gieras [71] also believes that when students believe in their ability to succeed and feel positive about their studies or success, their self-confidence rises and so does their motivation to learn. The personal connection with the teacher, emotionally supportive discourse, and positive reinforcements strengthen their perception that their success depends on them and their investment.

## 5. Discussion

Analysis of teachers’ reports reveals two central perceptions about literacy resilience: (1) literacy resilience is a tool for life, and (2) home literacy makes an important contribution to fostering literacy resilience. The concept of resilience describes optimal functioning and refers to abilities and skills that predict and build resilience in challenges and situations of uncertainty. It also includes the ability to overcome them, recover, and grow from them. The literacy resilience concept has become practical in teachers’ classroom work and focuses on students’ sense of capability. The participants revealed a pedagogical perception that building literacy resilience will allow students to feel better and more comfortable when they are required to deal with language challenges, identify language skills they have acquired in the past, and enhance their strengths and motivation while adapting to diverse learning environments. These findings are consistent with Shapira’s [72] perception that the circle of capability concerning resilience refers to how we perceive our ability to perform a specific task and deal with various tasks. Therefore, it includes skills for managing pressures, dealing with anxiety, challenges, difficulties, and uncertainty.

According to Bandura [73], there is a connection between the perception of self-efficacy and expectations for results, which are an expression of future perception. In other words, there is a connection between literacy resilience and students’ perception of the future who believe in their ability to create the changes they desire, to see the continuity between their actions and future results; therefore, they have a stronger commitment to shaping their future. The greater the literacy resilience, the higher the sense of self-efficacy, and accordingly, the future perception is positive [74]. Researchers Denz-Penhey and Murdoch [75] also perceive personal resilience as self-efficacy to act in challenging situations with a sense that the skills required for successful coping are within reach, and things will work out in the best way that can be expected. Similarly, literacy resilience is personal resilience expressed in the learner’s mastery of literacy skills that allow them to deal with literacy tasks, develop critical thinking, identify difficulties, and know how to overcome them through various and relevant strategies and metacognitive actions.

From a critical perspective, it can be argued that the State of Israel is characterized by many deep educational gaps, which are reflected in its low achievements in various comparison tests relative to other OECD countries. Educational inequality originating from socioeconomic gaps leads in turn to the exacerbation of economic inequality, harming social mobility and equality of opportunity. Fostering literacy resilience among all students has significant value in reducing gaps and promoting educational equality [76]. Closing the literacy gap is critical because literacy is a basic skill required not only for academic success but also for successful civic, social, and economic participation [77,78]. Another perception revealed by the teachers is that home literacy makes an important contribution to fostering literacy resilience, teaching that teachers believe parents can promote children’s literacy skills through conscious and directed actions in spoken and written modes. The research literature [35,79] attaches great importance to parental involvement in their children’s education and achievement promotion. It has been proven that parents who maintain contact with teachers and the school, assist with academic tasks, volunteer for activities, and attend meetings positively influence their children’s educational-learning process.

The teachers also report that some parents struggle to assist with literacy tasks, and this fact directly affects the pace of progress and achievements of the children. This finding is in affinity with other studies [80,81] that examined the interaction between teachers and educators and found that parents from low socioeconomic status struggle to cooperate with teachers, mediate academic knowledge, and provide emotional support to children due to a lack of knowledge and material resources.

Fostering literacy resilience poses significant challenges for teachers, who are required to ensure that every student receives the support and opportunities necessary to learn, establish literacy resilience, and progress. Teachers’ reports indicate that the three main challenges are: (1) teaching in a heterogeneous classroom, (2) encouraging parental involvement, and (3) fostering independent learners.

To foster literacy resilience in a heterogeneous classroom, teachers need to be sensitive to the unique needs of each student and plan personalized teaching-learning processes. Creating a pleasant and supportive learning environment will promote the academic, emotional, and personal growth of every student in the class. The teaching is differential, referring to the learner’s status, educational content, teaching methods, and the interaction between the individual and society, allowing each student to learn regardless of ability gaps [82,83,84]. There is evidence for the effectiveness of differential teaching in achieving learning goals through methods such as group work and task selection by students [85].

One way to develop skills for independent learning is by incorporating peer dialogue practices. Dialogic learning emphasizes the value of providing opportunities for students to think and participate in building their knowledge. This teaching allows children, in practice, to develop cognitive skills through social interaction. Dialogic teaching is interpreted as a sustainable approach that facilitates students’ academic development and fosters skills of independent learning and critical thinking [86,87]. Dialogue between learners can facilitate children’s cognitive development through language. Students freely express their views and reflect on their own and others’ opinions, raise emotions and intuitions, and thus even deepen their understanding of the other’s perspective [88,89].

Another way to foster an independent learner is to maintain a routine of personal conversations and establish a personal connection that can promote the learning process and make the learning environment an inviting, supportive, and encouraging place. When a student feels a personal and supportive connection with the teacher, they feel safer to expose themselves and develop a sense of belonging and support within the learning environment. In this way, the teacher can better understand the student’s needs and adapt the learning to their specific personality and abilities. This can facilitate learning and increase the student’s success. According to McCullough et al. [89], the private meeting where the teacher, the responsible adult, expresses empathy and willingness to listen to the student can be meaningful because it exposes the student to a figure they will aspire to emulate. Moreover, the student can find comfort, hope, and a listening ear to help them cope with emotional and academic challenges.

In conclusion, the study’s importance is providing a broad picture of teachers’ perceptions and actions to foster literacy resilience in heterogeneous classrooms in elementary schools. A connection was found between the declaration level on educational perceptions and the actions performed in alignment with these perceptions. Despite the challenges, teachers establish a personal and trusted relationship with their students, encourage parents to be active in fostering literacy resilience while providing personal, emotional, and academic responses through varied learning experiences.

Furthermore, the research findings show that the differential teaching found to be complex for teachers is revealed to be beneficial, and teachers make efforts to adapt teaching-learning processes to differentiation and to arouse interest and motivation among their students. Moreover, the research findings create new and relevant knowledge that can serve teachers and policymakers regarding pedagogical principles for promoting students in elementary school. New and relevant knowledge may break through and penetrate even on non-emergency days into teaching-learning processes in a heterogeneous classroom and teacher training processes.

The present study acknowledges certain limitations, primarily due to its qualitative nature and relatively small sample size, which focused on the subjective experiences of female elementary school teachers in Israel. Notwithstanding these constraints, the research has yielded novel theoretical and practical insights regarding the significance of incorporating home literacy practices in fostering literacy resilience within heterogeneous classroom environments. These findings may have important implications for both policymakers involved in teacher education training and practicing educators. Specifically, the results underscore the need to enhance awareness and sensitivity towards diverse family literacy contexts, potentially contributing to the development of more effective strategies for establishing and reinforcing literacy resilience among students.

## Data Availability

Data are unavailable due to privacy or ethical restrictions.

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
