# Peer review of "Beyond Magic: Fostering Literacy Resilience in Diverse Classrooms through Home-Based Approaches"

_behavsci, 2024, doi:10.3390/bs14090834_

Round 1

Reviewer 1 Report

Comments and Suggestions for Authors

Dear Author,

You have presented an interesting paper and you write comprehensibly, yet I did not understand the message of the paper. It starts at the very beginning, with the title of your paper. To summarize, your paper needs a clear thread that leads to the research questions and provides all the information the reader needs to understand your arguments, study approach, conclusions, etc. I felt overwhelmed by the amount of information you presented and felt that I was not prepared to tackle your paper. I will explain a few points below.

(1) Title: The title suggests a clear and exciting finding from your study, but after reading your summary, any reader will know that only teachers from a very narrow segment of education have spoken about their perceptions of resilience in relation to literacy skills in heterogeneous classrooms. I would prefer a title that more adequately represents your study.

(2) The summaries seem to be ok, but may no longer adequately reflect the work after revision.

(3) The paper starts very harshly by confronting the reader directly with the literature review on resilience. I would recommend a softer and more engaging introduction that reflects the motivation for your study. What is the problem you are focusing on? Is it paired with a specific research gap? How does it help with this problem if we know what teachers think about it? At the end, you can tell us the plan of your work to solve the problem (perhaps focusing only on the theoretical part of the work; in other words, what we should know in order to understand your study). This introduction should not be more than half a page, a third might be appropriate, but it depends on the message (content).

(4) I would like to read a focused theoretical part of the paper. Your research questions are related to (a) promoting literacy and integrating home literacy into classroom language instruction and (b) the challenges and coping methods related to (a). Therefore, as a reader, I would like to know what the state of knowledge is regarding (a) and (b). I have the feeling that I am not sufficiently informed. In connection with this gap in the reader's knowledge, I also do not understand why interviews are necessary in a qualitative study and how they go beyond our knowledge.

Yes, lines 115-123 in particular give a reason why you are targeting perceptions, but I can't follow your arguments. You write in lines 111-113: "Family members play a central role in children's literacy development, and they constitute a rich social, cultural and linguistic resource that is often untapped by the education system (Ishimaru et al., 2016)." But as you also indirectly state, families are not always this kind of resource that can be tapped into by the education system. However, what do we know about the available resources that we can find?

I feel like we get too much unfocused information in the theoretical part of your paper. I think we should learn more about "home learning environments in relation to literacy", perhaps about resilience pedagogy (line 87), perhaps about personalized learning and other learner-centered methods, etc. In this sense, I do not have the impression that the insights gained from the interviews are not already available in theory and research.

(5) The methods section could include more information about your interviews (structured, semi-structured, etc.) and the analysis. This is very general. Also, the transcripts are only analyzed by one person, so there is no check on the objectivity of the method. Also, the gender identity of the teachers is missing. Perhaps all the teachers are women?

(6) Results: In the "Results" section, some results are discussed and references are given. A section on results is only intended for results. Please move the discussion to the "Discussion" section and integrate it.

(7) Discussion: I think the discussion could mention again what we have gained from conducting a qualitative study. Be careful with over-generalizations of your results. 14 teachers in Israel is not the world, so there are limitations to your study that should be part of your discussion. A section on limitations is state of the art, but is missing from your paper.

Best regards

Reviewer

Author Response

Dear Reviewer,

Thank you for your important, accurate comments which have significantly contributed to the improvement of our article.

I carefully read all the comments and corrected the article accordingly.

All comments are highlighted in yellow in the body of the article.

Below is a summary of the key revisions:

(1) Title: The title suggests a clear and exciting finding from your study, but after reading your summary, any reader will know that only teachers from a very narrow segment of education have spoken about their perceptions of resilience in relation to literacy skills in heterogeneous classrooms. I would prefer a title that more adequately represents your study.

The title does not convey a specific finding. It does not suggest that teachers are successfully fostering literacy resilience in diverse classrooms. Furthermore, the phrase "Beyond Magic" might suggest the challenges and complexities teachers face in achieving this goal.

(2) The summaries seem to be ok, but may no longer adequately reflect the work after revision.

(3) The paper starts very harshly by confronting the reader directly with the literature review on resilience. I would recommend a softer and more engaging introduction that reflects the motivation for your study. What is the problem you are focusing on? Is it paired with a specific research gap? How does it help with this problem if we know what teachers think about it? At the end, you can tell us the plan of your work to solve the problem (perhaps focusing only on the theoretical part of the work; in other words, what we should know in order to understand your study). This introduction should not be more than half a page, a third might be appropriate, but it depends on the message (content).

As per your suggestion, I added an introduction that outlines the broader context of the significance of literacy resilience and the complex challenges that teachers face in addressing it.

(4) I would like to read a focused theoretical part of the paper. Your research questions are related to (a) promoting literacy and integrating home literacy into classroom language instruction and (b) the challenges and coping methods related to (a). Therefore, as a reader, I would like to know what the state of knowledge is regarding (a) and (b). I have the feeling that I am not sufficiently informed. In connection with this gap in the reader's knowledge, I also do not understand why interviews are necessary in a qualitative study and how they go beyond our knowledge.

Yes, lines 115-123 in particular give a reason why you are targeting perceptions, but I can't follow your arguments. You write in lines 111-113: "Family members play a central role in children's literacy development, and they constitute a rich social, cultural and linguistic resource that is often untapped by the education system (Ishimaru et al., 2016)." But as you also indirectly state, families are not always this kind of resource that can be tapped into by the education system. However, what do we know about the available resources that we can find?

I expanded the discussion and included additional information about the families that educators find challenging to motivate for involvement in fostering literacy resilience as part of the classroom learning process.

Educators often find it challenging to leverage the literacy resources available in students' homes, particularly when working with families from low socio-economic backgrounds who are marginalized socially. Parents may lack the knowledge, time, energy, language skills, transportation, flexible work schedules, or social support needed to effectively engage with their children's learning challenges (Lavenda, 2011; Hill and Tyson, 2009).

I feel like we get too much unfocused information in the theoretical part of your paper. I think we should learn more about "home learning environments in relation to literacy", perhaps about resilience pedagogy (line 87), perhaps about personalized learning and other learner-centered methods, etc. In this sense, I do not have the impression that the insights gained from the interviews are not already available in theory and research.

I added more targeted information about resilience pedagogy, incorporating perspectives from both children and teachers.

1.2 Resilience Pedagogy

Benard [8] proposes that Resilience Pedagogy is an approach that views enhancing and deepening resilience as an integral part of a teacher's role within the teaching-learning process. He argues that one of the key responsibilities of educators is to build resilience in students, particularly adolescents, by developing their ability to integrate fragmented experiences resulting from failures or traumatic events. He emphasizes that to effectively foster resilience in their students, teachers must first cultivate their resilience. Both teachers and students who have developed resilience are better equipped to navigate uncertainties, unexpected situations, and social distress in their daily lives.

Research widely acknowledges that teachers are among the most influential factors in the lives of children from diverse social backgrounds [9]. Namka [10] highlights the crucial role schools play in children's lives, often serving as their primary source of support. Schools provide a space where children receive attention, empathetic listening, and emotional support from their teachers. However, most educators tend to prioritize the didactic aspect of teaching, viewing it as an essential component and often focus on rewarding children for academic achievements or expected behavior. In contrast, the personal component that allows for flexibility and focuses on nurturing students' strengths while promoting well-being and resilience is often perceived as a less critical and potentially less effective aspect of education [11, 12].

(5) The methods section could include more information about your interviews (structured, semi-structured, etc.) and the analysis. This is very general. Also, the transcripts are only analyzed by one person, so there is no check on the objectivity of the method. Also, the gender identity of the teachers is missing. Perhaps all the teachers are women?

A more detailed description of the interviews is in the Methods section. I indicated that all teachers are women.

To address the research questions semi-structured interviews were conducted to efficiently engage participants in conversation and elicit their understanding and interpretation of a topic [36]. The interview was conducted face-to-face or by phone, lasting about 45 minutes each.

A more detailed description of the content analysis process is in the Methods section.

Content analysis facilitates an accurate data description and allows for drawing valid conclusions within a broader context. The process of coding and classifying findings is a cyclical task that bridges theory and content, requiring the researcher to dedicate time for reading, probing, refining, and decision-making.

The content analysis process consists of three main stages: (1) Open Coding: This initial stage involves defining the first categories of analysis. The researcher searches for recurring phrases and ideas, uncovering a set of topics and organizing them into central categories that reflect teachers' perceptions and pedagogical actions regarding the cultivation of literacy resilience and the integration of home literacy in Hebrew lessons. (2) Axial Coding: This stage involves rereading the texts to identify and refine the main thematic units. The researcher aims to tighten and determine which thematic unit each section belongs to, guided by the research goals and the theoretical conceptual framework that embodies the research objective. This framework focuses on cultivating literacy resilience in diverse classrooms while considering home literacy. (3) Selective Coding: In this final stage, the researcher determines recurring themes that will serve as central themes and adapts the segments to these themes. The goal is to develop a system of categories that provides a meaningful structure to the collected data. This approach aims to create a robust analytical framework that aligns with the research objectives and provides insightful interpretations of the data [40,41].

(6) Results: In the "Results" section, some results are discussed and references are given. A section on results is only intended for results. Please move the discussion to the "Discussion" section and integrate it.

My experience, along with my review of other articles and the comments of the second reviewer suggests that it is sometimes possible to broaden the interpretation by drawing on additional theoretical sources. For this reason, I chose to retain the extensive interpretation.

(7) Discussion: I think the discussion could mention again what we have gained from conducting a qualitative study. Be careful with over-generalizations of your results. 14 teachers in Israel is not the world, so there are limitations to your study that should be part of your discussion. A section on limitations is state of the art, but is missing from your paper.

As per your suggestion, I added a paragraph that describes the study's limitations.

The present study acknowledges certain limitations, primarily due to its qualitative nature and relatively small sample size, which focused on the subjective experiences of female elementary school teachers in Israel. Notwithstanding these constraints, the research has yielded novel theoretical and practical insights regarding the significance of incorporating home literacy practices in fostering literacy resilience within heterogeneous classroom environments. These findings may have important implications for both policymakers involved in teacher education training and practicing educators. Specifically, the results underscore the need to enhance awareness and sensitivity towards diverse family literacy contexts, potentially contributing to the development of more effective strategies for establishing and reinforcing literacy resilience among students.

Best Regards,

The author

Reviewer 2 Report

Comments and Suggestions for Authors

 line 153: replace (Lincoln, Lynham & Guba, 2011) with (Lincoln et al., 2011)

Lines 228 to 237 can be reinforced by studies that defend the strength of autonomous learning and its consequent pedagogical scaffolding thanks to the mediation of digital tools (synchronous and asynchronous). Some studies such as http://hdl.handle.net/10045/61841 show the benefits in language acquisition through this methodology.

It is clear from lines 251 to 290 that an analogy can be drawn with the situation experienced by COVID19 a few years ago. One could include studies from the home learning perspective focusing on how home learning was managed during that time and the testimony of teachers. 

Lines 373-377 it can be understood that moving the educational framework to WhatsApp groups implies that the necessary digital structure was lacking to run a virtual learning campus through digital educational platforms such as Moodle, Canvas, or others. 

As for the theoretical and bibliographical framework, it should be noted that almost all references are more than 10 years old. Some references from the last 5 years should be added, since, after the COVID pandemic, home learning has led to a transformation of the teaching and learning processes. 

Author Response

Dear Reviewer,

Thank you for your important, accurate comments which have significantly contributed to the improvement of our article.

I carefully read all the comments and corrected the article accordingly.

All comments are highlighted in yellow in the body of the article.

Below is a summary of the key revisions:

 line 153: replace (Lincoln, Lynham & Guba, 2011) with (Lincoln et al., 2011)

I have corrected and revised all the references to conform to the journal's accepted style.

Lines 228 to 237 can be reinforced by studies that defend the strength of autonomous learning and its consequent pedagogical scaffolding thanks to the mediation of digital tools (synchronous and asynchronous). Some studies such as http://hdl.handle.net/10045/61841 show the benefits in language acquisition through this methodology.

I expanded the discussion regarding autonomous learning mediation digital tools and reinforced it with references to new studies.

Studies indicate that a pedagogical setting based on synchronous and asynchronous digital tools can foster independent learning skills and promote active learning by encouraging learners to take responsibility for their education. This involves making informed choices that require a high level of awareness and control over the learning process [42]. Furthermore, motivation for autonomous learning is linked to positive self-experiences, fostering a sense of optimality and promoting optimal functioning [43].

It is clear from lines 251 to 290 that an analogy can be drawn with the situation experienced by COVID19 a few years ago. One could include studies from the home learning perspective focusing on how home learning was managed during that time and the testimony of teachers. 

As per your suggestion, I expanded the interpretation and added a paragraph based on studies from home learning, that draws an analogy between COVID-19 and literacy resilience concerning parental support and involvement.

Shmuel [44] found that teachers in the public education system in Israel are still not sufficiently aware of the importance of acquiring a deep understanding of their students' historical and cultural background, and still fail to build a true partnership with parents.

From this finding, it is possible to draw an analogy to studies that examined the involvement of parents in their children's learning process during the COVID-19 period. These studies found that the pandemic increased the need for frequent communication between teachers, parents, and students, as well as heightened the involvement of parents, family members, and the community [45,46,47]. Other studies [48,49,50] revealed that, from the parents' perspective, they enriched their understanding of learning processes, had the opportunity to assist their children in completing academic tasks, played a more significant role in their children's education, and developed a greater appreciation for the work of teachers. Similarly, a Portuguese study [51] demonstrated that parents adopted new interaction patterns to support their children, offering guidance, assistance, and continuous monitoring of educational task performance.

Lines 373-377 it can be understood that moving the educational framework to WhatsApp groups implies that the necessary digital structure was lacking to run a virtual learning campus through digital educational platforms such as Moodle, Canvas, or others. 

I expanded the interpretation of the lack of a digital structure to operate a virtual learning campus in the educational setting.

It appears that the educational setting lacks a robust technological-digital infrastructure necessary for the effective operation of a virtual learning campus through platforms like Moodle or Canvas. Evidence suggests that many educational institutions face significant challenges in implementing digital learning, primarily due to concerns regarding its effectiveness and the capacity of both students and educators to adapt to new digital learning environments. A major challenge is the requirement for high levels of self-discipline among learners, coupled with existing gaps in digital literacy skills and infrastructure [59, 60].

As for the theoretical and bibliographical framework, it should be noted that almost all references are more than 10 years old. Some references from the last 5 years should be added, since, after the COVID pandemic, home learning has led to a transformation of the teaching and learning processes. 

I added new references from the last five years to the theoretical and bibliographical framework.

Best Regards,

The author

Round 2

Reviewer 1 Report

Comments and Suggestions for Authors

Dear author,

I still think that a clear thread to the research questions could make the reading more comprehensible. You give a lot of information, but an academic reader should be able to cope with this challenge. Overall, the revision was worthwhile, especially the introduction to the article is now more adaptive. But please add some literature sources to the first four paragraphs.

Contrary to my objection to only reporting results in the results section, I think it is actually better that you have expanded the reference to the literature. In view of the complexity of your paper, I think this reduces the processing burden on the reader.

The limitations in the discussion section also put the rather strong conclusions into perspective and make them appear more moderate. In my opinion, the academic readership can also cope with this.

But please add some literature sources to the first four paragraphs.

Thank you for your very interesting article.

Best regards,

Reviewer 

Author Response

9/2/2024

Dear Reviewer,

I appreciate you taking the time to review my work and provide constructive input. Your comments have helped strengthen the paper, and I'm grateful for the opportunity to improve it based on your expertise.

I still think that a clear thread to the research questions could make the reading more comprehensible. You give a lot of information, but an academic reader should be able to cope with this challenge. Overall, the revision was worthwhile, especially the introduction to the article is now more adaptive. But please add some literature sources to the first four paragraphs.

I've incorporated the suggestions you made and updated the first four paragraphs accordingly. The changes are marked in yellow for the reference.

Best regards,

Author
